# Reformulating lipid nanoparticles for organ-targeted mRNA accumulation and translation

Kexin Su [1], Lu Shi[1,2], Tao Sheng[1], Xinxin Yan[1], Lixin Lin[1], Chaoyang Meng[3], Shiqi Wu[1,2], Yuxuan Chen [1], Yao Zhang[1], Chaorong Wang[1], Zichuan Wang[1], Junjie Qiu[1], Jiahui Zhao[1], Tengfei Xu[1], Yuan Ping [1,2] ✉, Zhen Gu [1,2] ✉ & Shuai Liu [1,2,4] ✉

Fully targeted mRNA therapeutics necessitate simultaneous organ-specific accumulation and effective translation. Despite some progress, delivery systems are still unable to fully achieve this. Here, we reformulate lipid nanoparticles (LNPs) through adjustments in lipid material structures and compositions to systematically achieve the pulmonary and hepatic (respectively) targeted mRNA distribution and expression. A combinatorial library of degradable-core based ionizable cationic lipids is designed, following by optimisation of LNP compositions. Contrary to current LNP paradigms, our findings demonstrate that cholesterol and phospholipid are dispensable for LNP functionality. Specifically, cholesterol-removal addresses the persistent challenge of preventing nanoparticle accumulation in hepatic tissues. By modulating and simplifying intrinsic LNP components, concurrent mRNA accumulation and translation is achieved in the lung and liver, respectively. This targeting strategy is applicable to existing LNP systems with potential to expand the progress of precise mRNA therapy for diverse diseases.

mRNA therapeutics have demonstrated substantial potential for the treatment of a broad spectrum of protein-related diseases, with their potential applications closely associated with advancements in targeted delivery technology[1-5]. Veritable targeting necessitates the occurrence of simultaneous mRNA accumulation and translation in specific organs for curative functions and minimized side effects. However, reduction of off-organ accumulation remains formidably challenging with existing mRNA carriers. Current organ-specific delivery systems exclusively enable targeted mRNA expression, while do not adequately tackle persistent hepatic accumulation[6-8]. The desynchrony between accumulation and translation poses a significant obstacle to the clinical translation of precise mRNA drugs, promoting the introspection and redesign of current delivery technologies.

Lipid nanoparticles (LNPs) represent the most clinically advanced mRNA delivery systems, yet their efficacy is primarily constrained in systemic hepatocyte targeting and intramuscular vaccine administration[9-11]. To unlock the full potential of mRNA therapeutics across various diseases, a genuine requirement exists in veritable organ-targeted delivery. For instance, numerous endeavors have been made for lung-selective mRNA delivery, exemplified by selective organ targeting (SORT) strategy[6,12,13], lipid structure adjustment in typical LNPs[8], and inhalation methodology[14,15]. Nevertheless, inhalation often leads to a low utilization rate of costly mRNA-loading nanoparticles. Despite the achievements in extrahepatic mRNA translation, hepatic accumulation of nanoparticles persists employing the two former approaches[6-8]. This off-organ

[1]State Key Laboratory of Advanced Drug Delivery and Release Systems, College of Pharmaceutical Sciences, Zhejiang University, Hangzhou, China. [2]Liangzhu Laboratory, Zhejiang University, Hangzhou, China. [3]Department of Hepatobiliary and Pancreatic Surgery, The First Affiliated Hospital, School of Medicine, Zhejiang University, Hangzhou, China. [4]Eye Center, The Second Affiliated Hospital, School of Medicine, Zhejiang University, Hangzhou, China. ✉e-mail: pingy@zju.edu.cn; guzhen@zju.edu.cn; shuailiu@zju.edu.cn

distribution prompts the reassessment of prevailing LNP systems for genuine targeting performance.

Current LNPs typically consist of four components: ionizable lipid/phospholipid/cholesterol/polyethylene glycol-lipid (PEG-lipid), while this stationary philosophy immensely contributes to similar physicochemical properties and inevitable liver tropism following systemic administration[9,16–18]. Notably, cholesterol existence in LNPs has been documented to facilitate lipoprotein coating, thereby enhancing interaction with hepatic cells[19,20]. Moreover, phospholipid contributes to the structural stability and endosomal escape process of formulations[21,22]. Breaking through current LNP paradigms, we discover that cholesterol and phospholipid are non-essential for LNP functionality. From these considerations, reformulating LNPs from dual constituents and varied chemical structures presents a promising avenue for veritable targeted delivery, enabling simultaneous mRNA accumulation and translation in organs of interest.

Here, we reformulate LNPs through lipid chemical structures and composition adjustments, enabling coinstantaneous mRNA accumulation and translation in the lung and liver, respectively. Unlike conventional cationic lipids derived from amine cores[23–25], a library of ionizable cationic lipids with biodegradable ester-cores (nAcx-Cm) is designed. nAcx-Cm LNPs with typical compositions mediate liver-specific mRNA accumulation and translation. In contrast to previous LNPs that primarily targeted hepatocytes, nAcx-Cm LNPs predominantly mediate mRNA accumulation and expression in hepatic endothelial cell subsets, addressing the challenging cell type-selectivity and expanding their applicability of disease range. Following this achievement, we discover that some inherent components of LNPs, such as cholesterol and phospholipid, are not indispensable for delivery, which instead lead to inevitable liver accumulation. Encouragingly, nAcx-Cm lipids are functioned synergistically with permanently cationic lipids and PEG-lipid for simultaneous pulmonary mRNA accumulation and translation, primarily in endothelial cells and epithelial cells. These three-component LNPs outperform their cholesterol-containing five- or four-component counterparts in both efficacy and veritable targeting property for lung delivery, simultaneously maintaining remarkable stability. Moreover, this lung-targeted formulation strategy showcases universality to other existing cationic lipids and LNPs. This methodological redesign of LNPs for veritable targeting presents significant clinical potential, establishing a standard for future targeted delivery.

## Results

### Rational design of degradable-core based ionizable cationic lipids for mRNA delivery

Advancements in material chemistry have propelled the development of various mRNA delivery systems, wherein LNPs stand out and have been employed in numerous clinical trials[3,26]. Ionizable cationic lipids represent the crucial component in LNPs, and extensive efforts have been made to design efficacious lipid structures[11,27,28]. Most current cationic lipid architectures typically originate from amines, featuring an amine-linker-tail formula[21,26,29–31]. The leading ALC-0315 and SM-102 lipids in FDA-approved mRNA vaccines further incorporated ester groups in the linker module for safety purpose[21]. However, even if the linkers may cleave, only hydrophobic tails are detached from the ionizable lipids, leaving behind amine and residues in the resulting molecules. Conversely, comprehensive core degradation of lipids will lead to smaller moieties, thereby facilitating LNP dissociation, mRNA release, and safety profiles. Based on these findings, we hypothesize that altering the intrinsic formula and introducing degradable cores into cationic lipids might provide a way forward for the development of next-generation lipid vectors with high efficacy and safety.

From the chemical perspective, a degradable core-amine-tail formula was designed and a combinatorial library of ester-core based ionizable cationic lipids (nAcx-Cm) was synthesized via a facile solvent-free one-pot approach, where n indicated the number of ester groups in one molecule (Fig. 1a, b). nAcx denoted distinct acrylic ester cores, and m referred to different hydrophobic alkyl chains in the lipids. Michael reaction of 14 degradable cores and 10 hydrophobic tail molecules yielded 140 nAcx-Cm lipids with controlled numbers of ester groups and hydrophobic tails, as well as variable chain lengths and branches (Fig. 1c, d and Supplementary Figs. 1, 2). The liver tropism of existing LNPs is derived from the internal constituents, and targeted mRNA accumulation and translation necessitate the redesign of current LNP formulation principles. These numerous cationic lipid architectures serve as the groundwork for subsequent LNP formulations, offering a versatile platform to explore varied compositions and lipid structures for veritable organ-targeted mRNA delivery (Fig. 1e, f).

### nAcx-Cm LNP screening for enhanced mRNA delivery

The transfection capacity is intricately linked to the vector chemical structures and nano-formulations[26,32–36]. An orthogonal assay was conducted in the first round to optimize the LNP formulations containing nAcx-Cm, 1,2-dioleoyl-sn-glycero-3-phosphoethanolamine (DOPE), cholesterol, and 1,2-dimyristoyl-rac-glycero-3-methoxy (poly(ethylene glycol-2000)) (DMG-PEG2000) in an ovarian cancer cell line (IGROV1) (Fig. 2a and Supplementary Figs. 3, 4). Afterwards, selected formulations underwent a second-round in vivo optimization to determine the top B-2 formulation with a nAcx-Cm/DOPE/cholesterol/DMG-PEG2000 molar ratio of 15/20/25/2 (Fig. 2b, c). The B-2 formulation ratio was used in the following nAcx-Cm screening, and nAcx-Cm LNPs mediated mRNA expression in a dose dependent manner (Fig. 2d and Supplementary Figs. 5, 6). Following a structure-activity relationship (SAR) analysis, the heat map illustrated the significant effect of branches and tail species in nAcx-Cm lipids on LNP-mRNA performance (Fig. 2e, f). nAcx-Cm lipids with multiple branched chains (4 ~ 6) outperformed their counterparts with fewer branches (1 ~ 3) (Fig. 2g). Moreover, lipids with a single hydrophobic tail on the branches exhibited higher mRNA delivery efficacy compared to those with double tails (Fig. 2h). Among these multiple-branched and single-tailed lipids, the hit rate reached 100% when RLU was counted over 100,000 (Fig. 2i). Additionally, all these LNPs showed low cytotoxicity (Supplementary Fig. 7). Further hydrolysis experiments combined with $^1$H NMR analysis confirmed the degradability of the designed ionizable lipids (Supplementary Fig. 8). The robust mRNA expression and favorable cellular tolerance endow nAcx-Cm lipids substantial potential for in vivo mRNA delivery applications, laying foundation for further targeted exploration.

### nAcx-Cm LNPs addressed multiple physiological barriers during mRNA delivery

Although numerous LNPs have been designed, most demonstrated limited capacity for endosomal escape[37,38]. Additionally, stability and mRNA release profiles should be addressed for enhanced delivery[39]. The nAcx-Cm LNPs showed particle sizes around 100 nm, and encouragingly, top 6Ac1-C12 LNPs remained stable after a 30-d incubation at 4 °C, holding potential to alleviate the challenge of cold chain transport of mRNA drugs (Fig. 3a, b). In contrast, the 6Ac1-C212 LNPs dissociated during the 30-d incubation, highlighting the advantageous role of single-tailed branches in retaining stability (Fig. 3c). Furthermore, nAcx-Cm LNPs exhibited slightly negative surface charges for serum resistance and appropriate mRNA binding efficacy (Supplementary Fig. 9).

Next, endosomal escape functionality was assessed, which had been regarded as the daunting step for RNA delivery[7,40,41]. Specifically, a fluorescence resonance energy transfer (FRET) assay was employed to evaluate the nAcx-Cm lipid fusion and disruption of endosomal membranes[7,25]. Two FRET probes, 1,2-dioleoyl-sn-glycero-3-phosphoethanolamine-N-(lissamine rhodamine B sulfonyl) (Rho-PE) and 1,2-dioleoyl-sn-glycero-3-phosphoethanolamine-N-(7-nitro-2-1,3-benzoxadiazol-4-yl) (NBD-PE), were incorporated into the

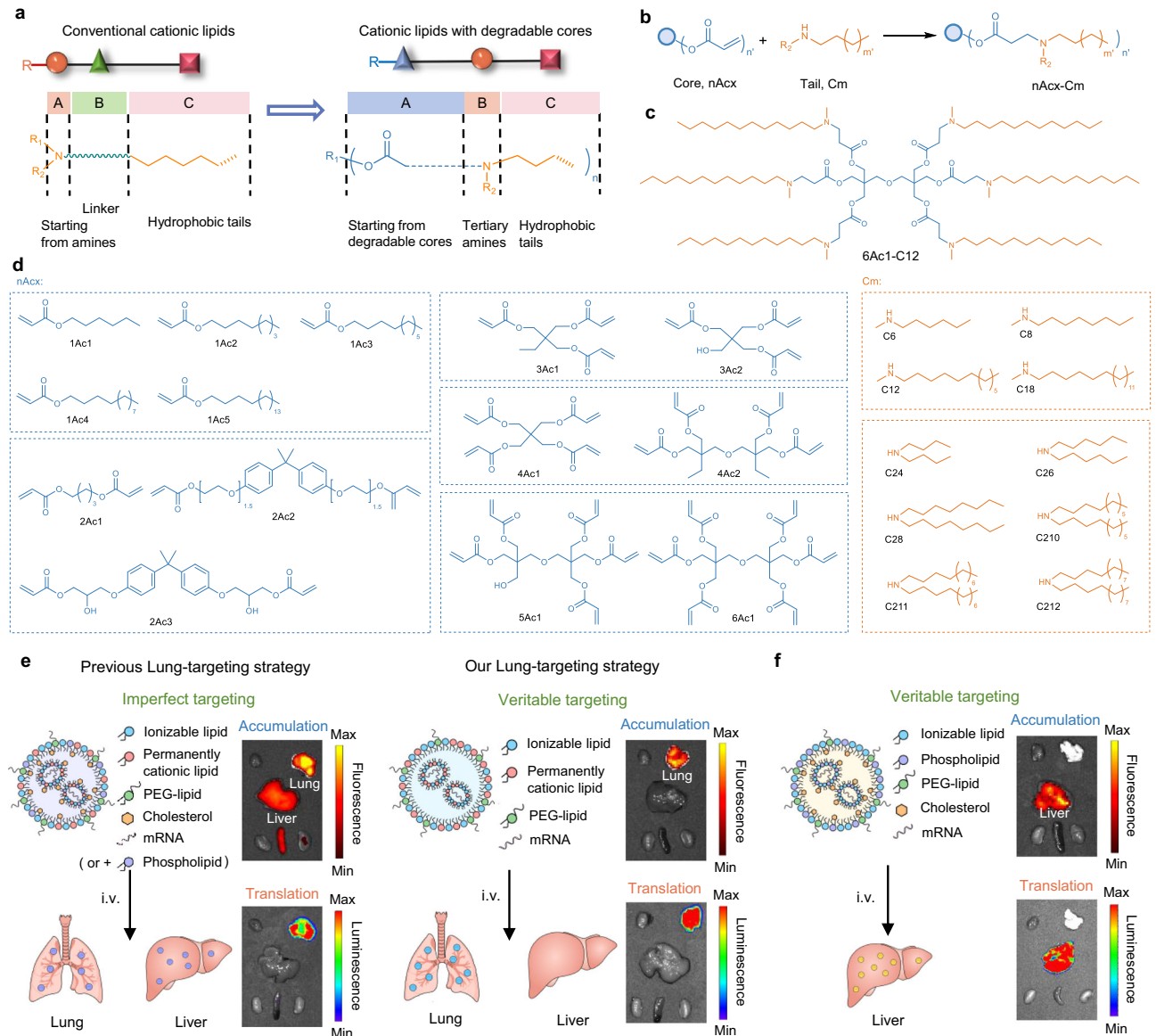

**Fig. 1 | A combinatorial library of ester core-amine-tail ionizable lipids was designed for mRNA targeted delivery. a** Unlike conventional cationic lipids starting from amines, an ester core-amine-tail formula was designed for cationic lipid synthesis. **b** Synthetic routes of nAcx-Cm lipids by Michael addition of degradable ester cores (nAcx) and secondary amines (Cm). n in nAcx-Cm indicates the number of ester bonds in one molecule. **c** Chemical structure of the representative 6Ac1-C12 lipid. **d** 14 degradable cores and 10 hydrophobic amines for lipid synthesis. Schematic representation of veritable organ-targeted mRNA accumulation and translation following systemic administration. Authentic lung- (**e**) and liver-targeting (**f**) was achieved by dually altering compositions and the lipid structures of LNPs. Representative mRNA accumulation and translation images from 6Ac1-C12 4-Comp Lung LNPs (imperfect lung targeting), 6Ac1-C12 3-Comp Lung LNPs (veritable lung targeting), and 6Ac1-C12 Liver LNPs (conventional compositions, veritable liver targeting) were presented, respectively.

endosomal mimicking liposomes, resulting in diminished NBD signal due to FRET to rhodamine (Fig. 3d). Upon lipid fusion, the increased distance between the two probes would lead to an augmented NBD signal. Notably, 6Ac1-C12 LNPs mediated higher membrane fusion than 6Ac1-C212 LNPs, revealing that the single-tail grafting on the branches facilitated the endosomal membrane rupture compared to the double-tail counterparts (Fig. 3e). Furthermore, membrane fusion of 6Ac1-C12 LNPs barely happened under physiological conditions, ensuring their biocompatibility for following in vivo utility (Fig. 3f). Thereafter, endocytosis and endosomal escape of 6Ac1-C12 LNPs were visually observed, evidenced by the Cy5-mRNA fluorescence images (Fig. 3g). Subsequently, mRNA should be released from nanoparticles for translation into proteins. 6Ac1-C12 LNPs were disassembled and mRNA were released efficiently once mixing with membrane mimics (Fig. 3h, i). Additionally, nAcx-Cm LNPs exhibited

p$K_a$ values around 6.0 (Fig. 3j). These characteristics suggest that nAcx-Cm LNPs possess excellent stability before cellular entry, and mRNA can be rapidly escaped from endosomes and released into the cytoplasm for subsequent translation (Fig. 3b–i).

## Lipid structures and components of nAcx-Cm LNPs controlled organ-targeted mRNA translation

Despite the substantial potential of mRNA therapeutics, their full realization across various diseases necessitates further advancements in targeted delivery technology. Given the intricate physiological environment in vivo, the exploration of targeted mRNA delivery systems remains challenging[42–44]. Notably, tailoring vector structures emerges as a promising avenue for enhanced efficacy and targeting. We firstly selected 11 nAcx-Cm lipids from the in vitro screening and evaluated their capacities of in vivo performance (Fig. 4a). As expected, lipid

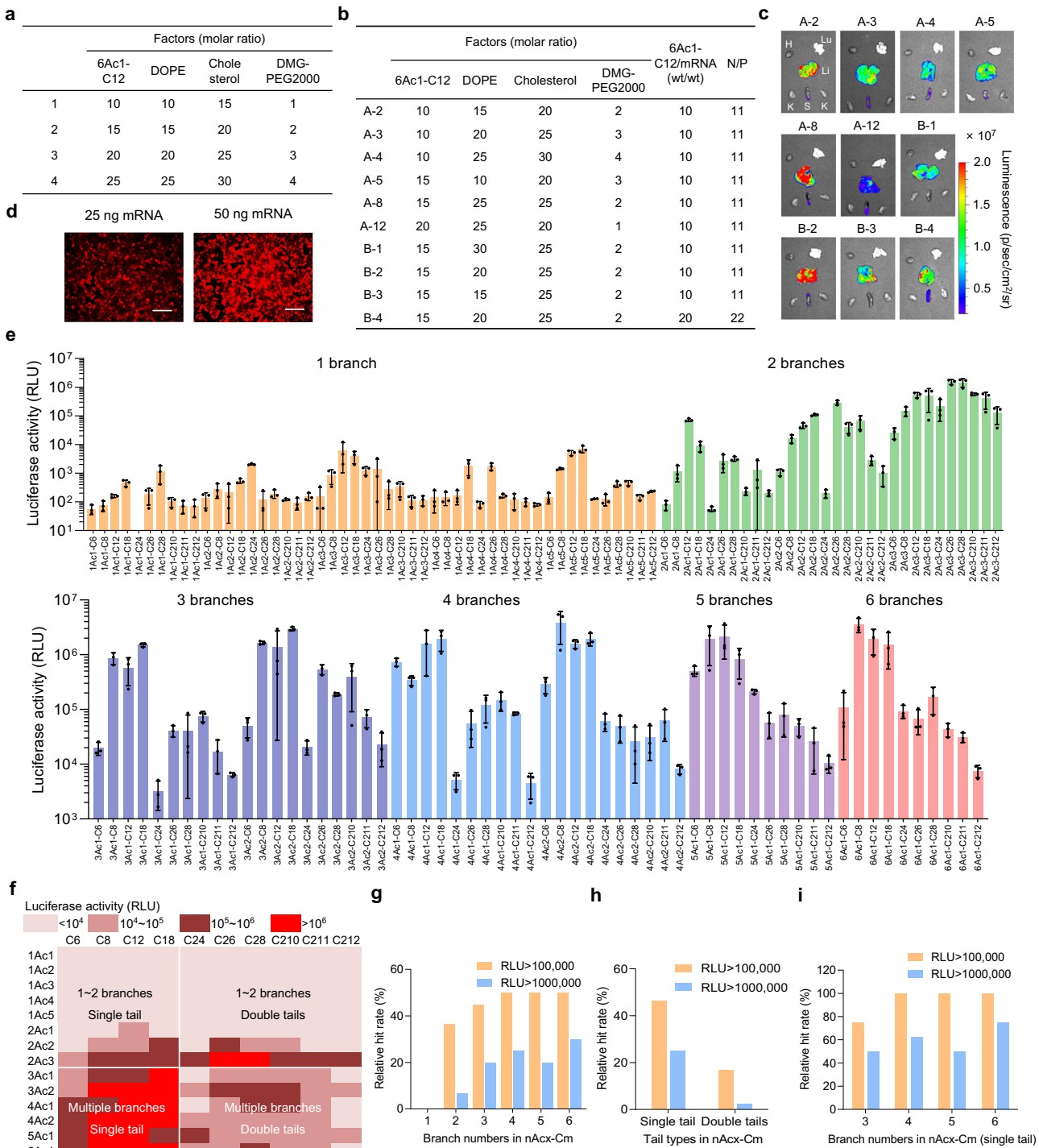

**Fig. 2 | LNP optimization and structure-activity relationship (SAR) study of cationic lipids for mRNA delivery. a** Orthogonal screening of LNP formulations with four levels. **b** LNP formulation table for mRNA delivery validation in vivo. **c** Firefly luciferase (Fluc) mRNA translation by different LNP formulations (0.25 mg kg[-1] mRNA). H heart, Lu lung, Li liver, K kidney, S spleen. **d** 6Ac1-C12 LNPs delivered mCherry mRNA at different mRNA doses. Scale bar: 200 μm. **e** In vitro screening of nAcx-Cm lipids by Fluc mRNA delivery to IGROV1 cells, recorded by relative light units (RLU) (25 ng mRNA, $n = 3$ biologically independent samples). **f** The heat map of luciferase expression upon transfecting IGROV1 cells with nAcx-Cm LNPs. **g** Relative hit rate of nAcx-Cm lipids with different branch numbers. **h** Relative hit rate of nAcx-Cm lipids with a single or double hydrophobic tails per branches. **i** Regarding nAcx-Cm lipids with the single tail per branch, 4-6 branched lipids exhibited high relative hit rate.

structures significantly affected the in vivo efficacy, with moderate tail lengths (around C12) facilitating high mRNA delivery efficacy (Supplementary Fig. 10). Moreover, the optimized LNPs exhibited even higher (or at least comparable) mRNA expression compared to FDA-approved DLin-MC3-DMA, SM-102, and ALC-0315 LNPs post intravenous (i.v.) or intramuscular administration (Fig. 4a, b and

Supplementary Fig. 11). The 6Ac1-C12 lipid with the six-ester core and six hydrophobic tails stood out, and the formulated LNPs enabled roughly 98% mRNA expression in the liver post systemic administration (Fig. 4c, d).

We next investigated the degradable core effect in organ-targeting (Supplementary Figs. 12 and 13). Remarkably, 2Ac3-C18

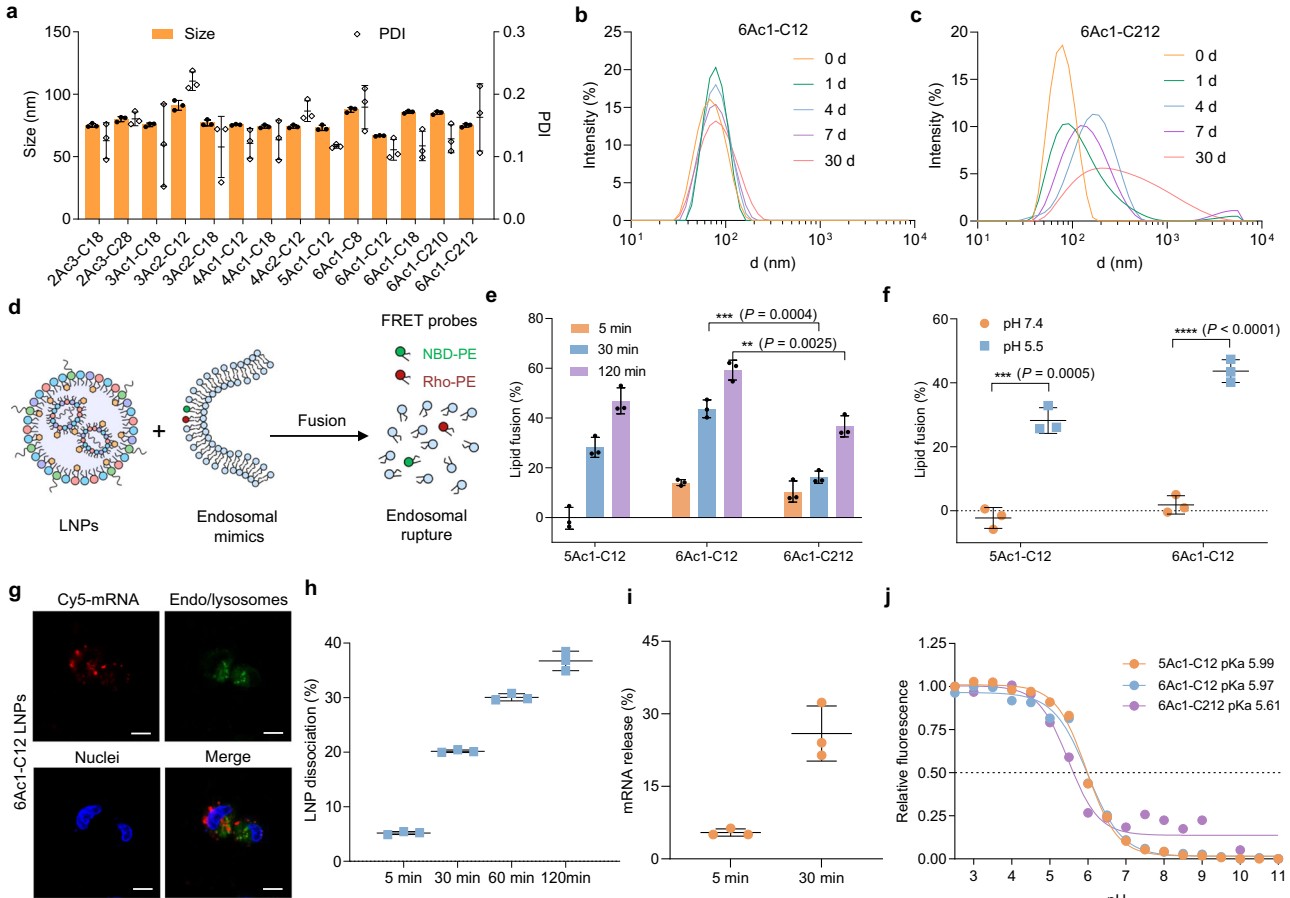

**Fig. 3 | nAcx-Cm LNPs demonstrated excellent stability, endosomal escape, and mRNA release functionalities. a** Particle size and polydispersity index (PDI) of nAcx-Cm LNPs. **b** 6Ac1-C12 LNPs remained stable after 30-d incubation at 4 °C. **c** 6Ac1-C212 LNPs exhibited instability as time prolonged. **d** A schematic showing the stimulation of membrane fusion and endosomal escape by the fluorescence resonance energy transfer (FRET) assay. **e** Assessment of lipid fusion and membrane rupture of nAcx-Cm LNPs by FRET assay at pH 5.5. Lipid fusion of LNPs were determined at different time intervals. **f** Membrane fusion of nAcx-Cm LNPs at different pHs. **g** Cellular uptake and endosomal escape fluorescence images of IGROV1 cells treated with Cy5-mRNA loaded 6Ac1-C12 LNPs. Scale bar: 10 μm. **h** Determination of 6Ac1-C12 LNP dissociation ability. **i** Evaluation of mRNA release capacity from 6Ac1-C12 LNPs. **j** The p$K_a$ of nAcx-Cm LNPs. Data are presented as mean±s.d. ($n$ = 3 biologically independent samples). Statistical significance in **e** and **f** was calculated using a two-tailed unpaired $t$-test: ****$P$ < 0.0001; ***$P$ < 0.001; **$P$ < 0.01; *$P$ < 0.05.

with two branches showed promising splenic targeting in vivo, which may be conducive to the future implementation of immunotherapy applications (Fig. 4e). Furthermore, the administration routes should also be taken into account for targeting exploration. Consistent with the previous organ-tropism[12,45], 6Ac1-C12 LNPs with the 1,2-dioleoyl-3-trimethylammonium-propane (DOTAP) helper lipid facilitated mRNA expression in the pancreas and lung post intraperitoneal and i.v. injection, respectively (Supplementary Fig. 14). These results demonstrate the feasibility of organ-selectivity achievement by adjustment in lipid structures or injection routes.

The inherent four LNP constitutes-ionizable lipid/phospholipid/cholesterol/PEG-lipid, have become the obstacles for mRNA delivery outside the liver[9,16,46]. Beyond vector structural and administration alteration, redesigning LNP components has attracted increasing attention for extrahepatic delivery. For instance, Siegwart group developed a SORT strategy and incorporating the fifth permanently cationic lipids into traditional LNPs induced lung-specific mRNA expression[12,13]. Alternatively, substituting phospholipids with permanently cationic lipids in four-component LNPs similarly led to lung-selectivity[47,48]. To further simplified the LNP systems and improve the lung-targeting functionality, we designed the three-component (3-Comp) ionizable cationic lipid/permanently cationic lipid/PEG-lipid LNPs for enhanced pulmonary delivery (Fig. 4f–h). Unlike conventional

LNP methodologies, we reported that phospholipid and cholesterol were removable from LNPs to optimize mRNA delivery, marking a bold innovation in LNP technology. Better still, the 3-Comp strategy exhibits versatility across various other LNP systems. Firstly, this approach was applicable to other ionizable cationic lipids for liver-to-lung transition, proved by SM-102, ALC-0315 and DLin-MC3-DMA 3-Comp LNPs (Supplementary Fig. 15). Secondly, employing different permanently helper lipids, such as DOTAP and (dimethyldioctadecylammonium bromide salt (DDAB), consistently retained the same lung-tropism (Fig. 4g, h). Meanwhile, we formulated more simplified two-component nAcx-Cm/PEG-lipid LNPs, failing to mediate mRNA expression in vivo (Supplementary Fig. 16).

Apart from above innovative 3-Comp LNP strategy, four-component (4-Comp) nAcx-Cm/cholesterol/permanently cationic lipid/PEG-lipid (Fig. 4i–k) and five-component (5-Comp, SORT strategy) nAcx-Cm/phospholipid/cholesterol/permanently cationic lipid/PEG-lipid (Fig. 4l–n) LNPs were further formulated, also mediating appropriate physicochemical properties (Supplementary Figs. 17, 18) and lung-selective mRNA expression (Fig. 4f–n and Supplementary Figs. 19–21). Notably, the simplified 3-Comp Lung LNPs facilitated higher efficacy compared to their 4-Comp or 5-Comp counterparts, highlighting the significance of LNP engineering innovation for superior efficacy and targeting (Supplementary Fig. 22). Given that

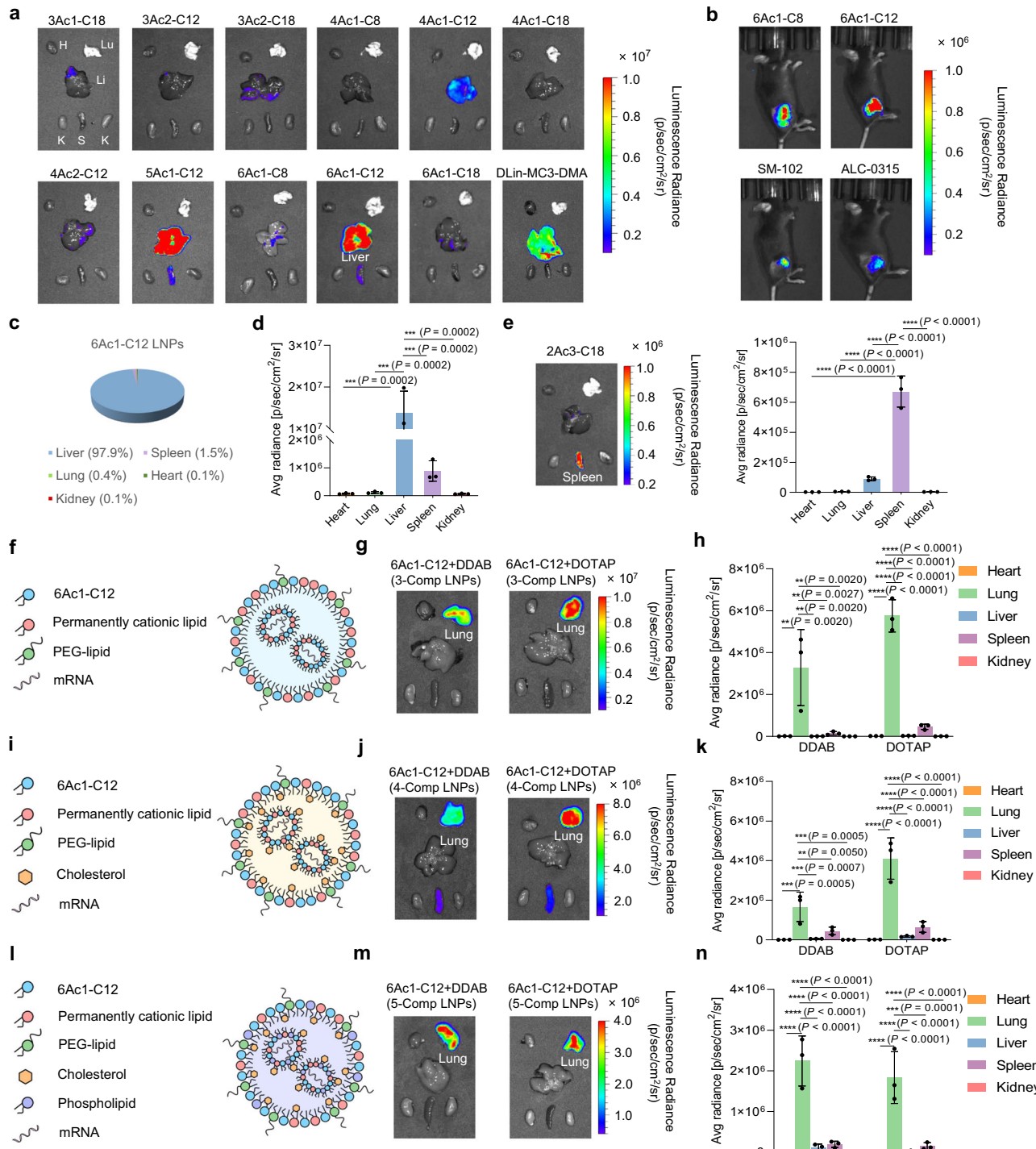

**Fig. 4 | Lipid structure and component adjustment in LNPs controlled organ-specific mRNA translation. a** In vivo mRNA delivery efficacy of nAcx-Cm LNPs. H heart, Lu lung, Li liver, K kidney, S spleen. **b** LNPs outperformed SM-102 and ALC-0315 LNPs in mRNA delivery post-intramuscular administration. 6Ac1-C12 LNPs induced liver-targeted mRNA delivery. Organ distribution (**c**) and quantification (**d**) of Fluc mRNA expression were assayed. Quantification was recorded by the average (Ave) radiance. **e** 2Ac3-C18 LNPs exhibited spleen-specific mRNA expression via systemic administration. 3-Comp nAcx-Cm/permanently cationic lipid/PEG-lipid LNPs without cholesterol and phospholipid mediated high lung-targeted mRNA delivery efficacy. Schematic (**f**), organ images (**g**), and quantification (**h**) of Fluc

mRNA translation by 3-Comp Lung LNPs were evaluated. 4-Comp nAcx-Cm/permanently cationic lipid/cholesterol/PEG-lipid LNPs mediated lung-selective mRNA translation. Schematic (**i**), in vivo evaluation (**j**), and quantification (**k**) of Fluc mRNA expression by 4-Comp Lung LNPs were recorded. 5-Comp nAcx-Cm/permanently cationic lipid/phospholipid/cholesterol/PEG-lipid LNPs induced mRNA delivery to the lung. Schematic (**l**), organ images (**m**), and quantification (**n**) of Fluc mRNA delivery by 5-Comp lung LNPs were shown. All data are presented as mean±s.d. (n = 3 biologically independent animals). All statistical significances were calculated using one-way ANOVA with Dunnett's multiple comparisons test: ****$P < 0.0001$; ***$P < 0.001$; **$P < 0.01$; *$P < 0.05$.

current LNPs can not circumvent the accumulation in liver due to the inherent components and physicochemical properties, the removal of original constitutes demonstrates substantial potential for achieving veritable organ-targeted mRNA accumulation and translation.

### Reformulating LNPs for simultaneous organ-targeted mRNA accumulation and translation

Targeted mRNA expression remains a persistent challenge, let alone simultaneous mRNA accumulation and translation in organs of interest. Although some progress has been made towards organ-selective mRNA expression, none has evaded the liver and nanoparticles are still biodistributed in the liver[6–8]. Addressing the accumulation issue is crucial for developing authentic organ-targeted delivery technology that meets clinical requirements with minimized side effects. We hypothesize that the liver tropism of LNPs derives from their intrinsic component-associated properties, and re-modulating the LNP constitutes may pave the way for veritable extrahepatic targeting.

Starting from the typical helper lipids comprising DOPE, cholesterol, and PEG-lipid, 6Ac1-C12 LNPs enabled liver-targeted accumulation and protein expression (Figs. 4c and 5a, b). In contrast to prior LNPs predominantly mediating hepatocyte uptake, the nAcx-Cm LNPs entered both endothelial and kupffer cells (Fig. 5c and Supplementary Fig. 23). To validate the cell type-specific mRNA translation, we utilized a Cre-LoxP mouse model containing Lox-Stop-Lox cassette to prevent tdTomato expression[7,29,49]. Upon delivering Cre-recombinase mRNA (Cre mRNA) in LNPs, the translated Cre protein could delete the stop cassette to activate tdTomato fluorescence only in transfected cells (Fig. 5d). Consistently, the mRNA expression of both 6Ac1-C12 Liver LNPs and 5Ac1-C12 Liver LNPs primarily occurred in endothelial cells, with approximated 60% transfection detected (Fig. 5e and Supplementary Fig. 24). These results demonstrate that tailoring cationic lipids can lead to cell subtype-targeting beyond hepatocytes, thereby addressing the treatment of diseases characterized by genetic abnormality in specific cell types.

Nanoparticles containing cholesterol typically facilitate lipoprotein coating in the bloodstream, showing pronounced affinity towards the liver[19,50]. We hypothesize that cholesterol-removal from LNPs might induce non-liver targeted mRNA accumulation and translation, addressing the formidable challenge of veritable extrahepatic targeting. We firstly revealed that eliminating cholesterol from 6Ac1-C12 Liver LNPs resulted in reduced mRNA expression in the liver, accompanied by increased delivery to the spleen, thereby diminishing liver targeting (Fig. 5f–h). As anticipated, the 3-Comp Lung LNPs without cholesterol not only facilitated lung-specific mRNA expression, but also demonstrated lung-targeted accumulation (Fig. 5i and Supplementary Fig. 25). This represents a milestone breakthrough for extrahepatic targeting, since previous lung-selective LNPs still partially accumulated in the liver[6–8]. Consequently, the 3-Comp Lung LNPs surpassed their 4-Comp and 5-Comp counterparts in both efficacy and targeting property (Fig. 5i, j and Supplementary Fig. 22).

Cholesterol is usually associated with lipoprotein coating and liver tropism[19,20], representing a pivotal element influencing the accumulation behavior of nanoparticles in vivo. Numerous prior efforts have focused on functionalizing cholesterol through chemical modifications or charge variations, enabling enhancement of mRNA-LNP delivery efficacy and accomplishment of extrahepatic mRNA expression[44,51–54]. In contrast, the 3-Comp Lung LNPs thoroughly eliminated the cholesterol here. To investigate the endogenous mechanism of specific targeting, we analyzed the protein coronas of different LNPs after incubation in plasma. Initially, 3-Comp Lung LNPs adsorbed less lipoproteins compared to Liver LNPs (Supplementary Fig. 26). Subsequently, to meticulously study the cholesterol effect on lung accumulation, 3-Comp and 4-Comp Lung LNPs were further selected for protein corona analysis, whose compositions differed only in the presence or absence of cholesterol. Figure 5k showed that

cholesterol removal from LNPs resulted in reduced lipoprotein coating, and this may explain the genuine lung targeting by the 3-Comp LNP methodology. Following these, this targeting strategy was validated on SM-102 lipid, demonstrating the superiority of lung targeted accumulation using 3-Comp LNPs (Supplementary Fig. 27). These mRNA loaded Lung LNPs accumulated in roughly 60% of all endothelial cells, with a less distribution in epithelial and immune cells (Fig. 5l and Supplementary Fig. 28). Moreover, the 3-Comp Lung LNPs mediated mRNA translation to 71% of all pulmonary endothelial cells, 34% of all epithelial cells, and 15% of immune cells (Fig. 5m and Supplementary Figs. 29, 30). It is noteworthy that cholesterol removal did not affect the nanoparticle stability, and all these LNPs retained unchanged sizes post 30-d storage (Fig. 5n and Supplementary Fig. 31). Moreover, these Liver- or Lung-targeted nAcx-Cm LNPs were well tolerated in vivo (Fig. 5o and Supplementary Figs. 32–34). These results establish the proposed 3-Comp Lung LNPs as a compelling candidate for the treatment of lung-specific diseases, such as pulmonary cystic fibrosis, with minimal side effects.

## Discussion

Recognized by the approval of COVID-19 mRNA vaccines and the Nobel Prize in Physiology or Medicine 2023, mRNA-based therapeutics are opening a new era for the medical industry[3,55,56]. Nevertheless, beyond vaccines, the progress of therapeutic mRNA drugs has been somewhat impeded by current incomplete targeting technology. Simultaneous organ-targeted mRNA accumulation and translation is still unattainable for extrahepatic tissues. Therefore, achieving veritable organ-targeted delivery will significantly propel advancements in the field of mRNA therapeutics, unlocking their full potential in treating a wide range of diseases.

We innovate the LNP delivery technology by eliminating cholesterol and phospholipid, retaining LNP functionality and endowing veritable targeting. LNPs are rationally engineered from lipid structures and components to achieve challenging simultaneous organ-targeted mRNA accumulation and translation. Thoroughly altering the LNP philosophy, 3-Comp and typical LNPs are developed based on the core-degradable cationic lipids, enabling veritable targeting to lung and liver, respectively. Unlike previous LNPs that target hepatocytes, tailoring nAcx-Cm lipids of typical LNPs induces hepatic endothelial cell selectivity. Following this, the 3-Comp LNP strategy removing cholesterol addresses the formidable challenge of evading liver accumulation, enabling authentic veritable pulmonary targeting (both accumulation and translation) following systemic administration. This organ-targeting methodology is applicable to other existing ionizable cationic lipids and LNPs, enriching the diversity of current LNP delivery systems and expanding their targeting possibilities. Our veritable organ-targeting methodology demonstrates great promise for the development of precise mRNA drugs with minimized side effects.

## Methods
### Materials

DOPE, 1,2-distearoyl-sn-glycero-3-phosphocholine (DSPC), DMG-PEG2000, DDAB, DOTAP (chloride salt), NBD-PE, and Rho-PE were purchased from Avanti Polar Lipids. Other chemicals were purchased from Sigma-Aldrich. Trypsin-EDTA (0.25%) and Penicillin-Streptomycin were purchased from Thermo Fisher. Dulbecco's modified phosphate buffered saline (PBS), RPMI 1640 medium, and fetal bovine serum were purchased from Sigma-Aldrich. The Bio-Lumi™ II Firefly Luciferase Assay Kit was purchased from Beyotime, and the alamarBlue kit was purchased from Yeasen. D-Luciferin firefly, sodium salt monohydrate was purchased from Gold Biothechnology. Fluc mRNA and Cy5-Fluc mRNA were purchased from Absin (Shanghai) Biotechnology Co., Ltd. mCherry mRNA and Cre mRNA was obtained from RNAlfa Biotech. Lipopolysaccharide (LPS) was purchased from Ranjco Technologies. LysC and trypsin enzymes were purchased from

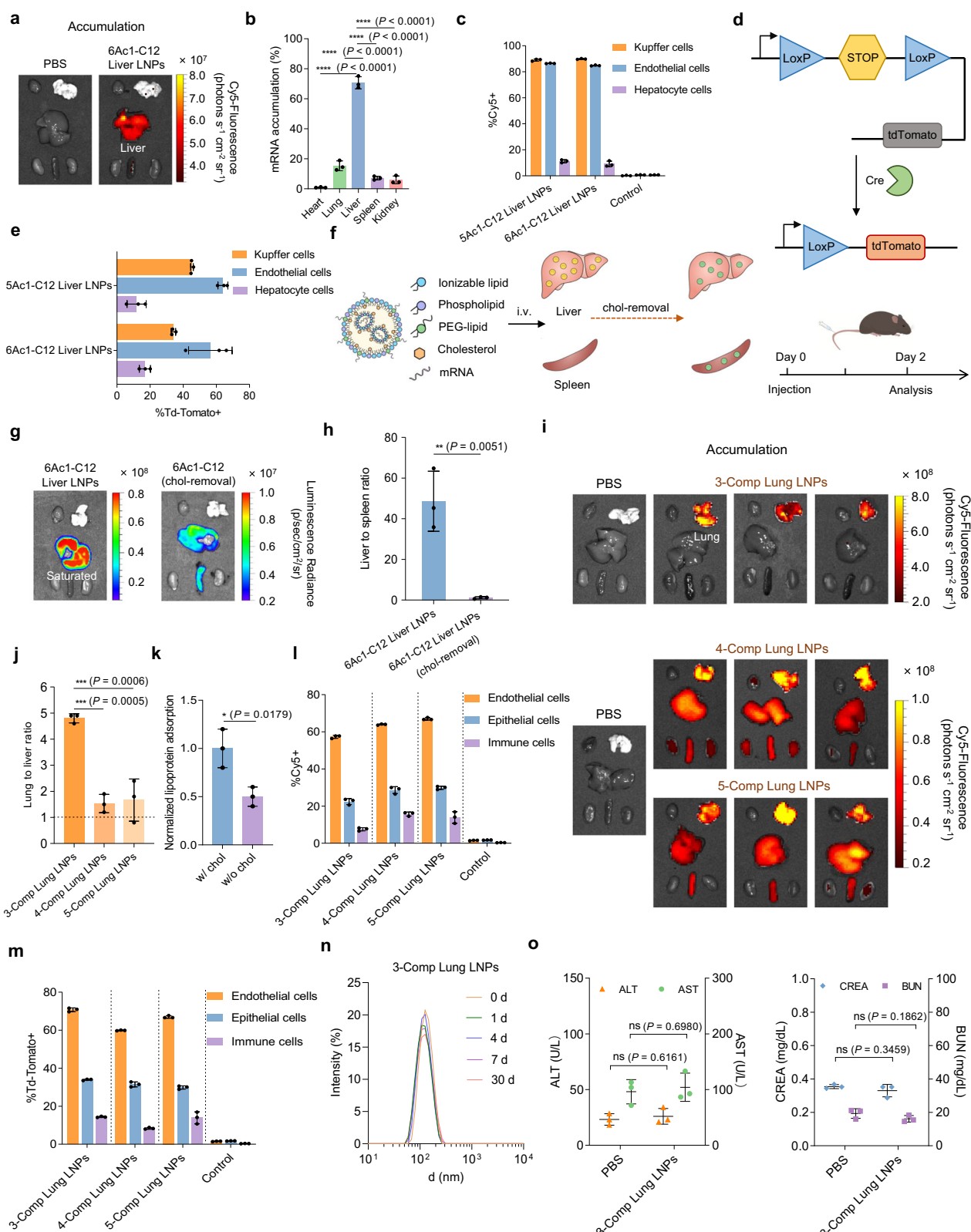

Shengxia Protein Technology Co., Ltd. The Quant-iT Ribogreen RNA assay kit was purchased from Life Technologies.

## Synthesis of core-degradable cationic lipid library

nAcx-Cm lipids were synthesized through Micheal addition reaction by acrylate monomers (nAcx) and amines (Cm). n indicated the number of degradable ester groups in one lipid molecule, and m determined the hydrophobic tail lengths and tail types. Equal equivalent of acrylate monomers were mixed with amines, and the mixtures were stirred at 60 °C for 72 h to give the crude products. Purified cationic lipids were obtained by silica gel column chromatography with the hexane/acetone elution, followed by rotary evaporation and vacuum drying.

To prepare the representative 6Ac1-C12 lipid, dipentaerythritol hexaacrylate (0.6 mmol) and N-methyldodecylamine (3.6 mmol) were

**Fig. 5 | LNP composition adjustment enabled simultaneous organ-targeted mRNA accumulation and translation. a, b** Typical four-component 6Ac1-C12 LNPs primarily accumulated in the liver post i.v. administration. Accumulation images (**a**) and quantification (**b**) of 6Ac1-C12 LNPs containing Cy5-mRNA were evaluated. **c** FACS was used to determine the mRNA distribution in hepatic cell types after treating 5Ac1-C12 and 6Ac1-C12 Liver LNPs. **d** The schematic showed the deletion of stop cassette by delivering Cre mRNA to Ai9 mice for activation of tdTomato expression. **e** Hepatic cell type transfection analysis of 5Ac1-C12 and 6Ac1-C12 Liver LNPs. Cholesterol removal from Liver LNPs decreased the liver specificity (Fluc mRNA, 0.25 mg kg⁻¹). Schematic (**f**), in vivo images (**g**), and targeting alteration (**h**) of 6Ac1-C12 Liver LNPs after removing the cholesterol component were shown. 3-Comp Lung LNPs without cholesterol and phospholipid demonstrated superior pulmonary targeting compared to 4- and 5-Comp Lung LNPs. The accumulation (**i**) and lung-targeting (**j**) of 6Ac1-C12 Lung LNPs with different components were assayed. **k** Cholesterol effect on lipoprotein adsorption of Lung LNPs by protein corona analysis. **l** Pulmonary cell type distribution analysis of Lung LNPs. **m** Quantification of pulmonary cell transfection by Lung LNPs. **n** Stability of 6Ac1-C12 3-Comp Lung LNPs. These LNPs remained stable post 30-d incubation. **o** 6Ac1-C12 3-Comp Lung LNPs were well tolerated in vivo. Alanine aminotransferase (ALT), aspartate aminotransferase (AST), blood urea nitrogen (BUN), and creatinine (CREA) were evaluated. All data are presented as mean±s.d. ($n$ = 3 biologically independent animals). Statistical significance in **h, k, o** was calculated with a two-tailed unpaired $t$-test, and those in **b, j** were calculated using one-way ANOVA with Dunnett's multiple comparisons test: $^{****}P < 0.0001$; $^{***}P < 0.001$; $^{**}P < 0.01$; $^{*}P < 0.05$; ns, no significant difference.

mixed and stirred at 60 °C for 72 h to yield the crude product. Subsequently, the product was purified by silica gel column chromatography eluting with hexane/acetone (4/1, v/v). After rotary evaporation and vacuum drying, purified 6Ac1-C12 lipid was obtained.

### nAcx-Cm LNP formulation and characterization
nAcx-Cm LNPs were formulated using the ethanol dilution method. Calculated nAcx-Cm lipids, cholesterol, phosholipid, and DMG-PEG2000 were dissolved in ethanol. mRNA was diluted in sodium citrate buffer (10 mM, pH 4.0). These two solutions were rapidly mixed at an ethanol to aqueous ratio of 1/3 by volume, and then LNPs were incubated at room temperature for 15 min. For in vivo experiments, LNPs were further dialysed against 1× PBS buffer in Pur-A-Lyzer chambers for 2 h.

### In vitro screening of nAcx-Cm LNPs for mRNA delivery
IGROV1 cells were cultured in RPMI-1640 medium with 10% FBS and 1% Penicillin/Streptomycin (P/S) at 37 °C/5% CO₂. In white 96-well plates, IGROV1 cells were seeded at the cell denstiy of $1.5 \times 10^4$ cells/well. 24 h later, Fluc mRNA loaded nAcx-Cm LNPs were prepared and added into the plates at a mRNA dose of 25 ng per well. The weight ratio of nAcx-Cm lipid/mRNA was fixed at 10/1, and nAcx-Cm lipid/DOPE/Cholesterol/DMG-PEG2000 molar ratio of 15/20/25/2 was used. Unless otherwise stated, this formulation ratio was employed for other physicochemical characterizations and in vivo screening assessments. For orthogonal assay, different weight ratios and molar ratios were assessed to optimize the LNP formulations. Luciferase expression and cell viability were assessed by the Bio-Lumi™ II Firefly Luciferase Assay Kits and alamarBlue kits, respectively. For mCherry mRNA delivery, IGROV1 cells were seeded in transparent 96-well plates. Post transfection for 24 h, fluorescence images of cells were captured by the inverted fluorescence microscope.

### Animal experiments
All animal protocols were approved by the Laboratory Animal Welfare and Ethics Committee of Zhejiang University (ZJU20220156). Normal wild-type C57BL/6 mice were purchased from GemPharmatech Co., Ltd. B6.Cg-*Gt(ROSA)26Sor^tm9(CAG-tdTomato)Hze*/J mice (Ai9 mice) were obtained from The Jackson Laboratory and bred to maintain the homozygous expression of Cre reporter allele that has a LoxP-flanked STOP cassette. Mice were maintained in a barrier facility with a 12-h light/12-h dark cycle, at around 20 °C and 40% humidity.

### In vivo mRNA delivery
An orthogonal assay was used to optimize the LNP formulations with different molar ratios. For the following in vivo nACx-Cm screening, ionizable lipids/DOPE/cholesterol/DMG-PEG2000 molar ratio was set as 15/20/25/2. DLin-MC3-DMA, SM-102, and ALC-0315 LNPs were set as the positive control. Briefly, DLin-MC3-DMA LNPs were formulated with a DLin-MC3-DMA/DSPC/cholesterol/DMG-PEG2000 molar ratio of 50/10/38.5/1.5. While for SM-102 and ALC-0315 LNPs, SM-102/DSPC/

cholesterol/DMG-PEG2000 molar ratio of 50/10/38.5/1.5 and ALC-0315/DSPC/cholesterol/ALC-0159 molar ratio of 46.3/9.4/42.7/1.6 were employed. LNPs were i.v. or intramuscularly injected to C57BL/6 mice. 6 h later, luciferase expression was assessed by the animal bioluminescence imaging. Mice were anaesthetised and 100 μL of D-luciferin substrate (30 mg mL⁻¹) was intraperitoneally administered. The body and isolated organ luminescences were quantified by the IVIS Spectrum system (Perkin Elmer).

For the lung-targeted LNP formulations, an orthogonal assay and rational design of component molar ratios were conducted. Lung delivery was conducted at a Fluc mRNA dosage of 0.25 mg kg⁻¹. Ultimately, 3-Comp Lung LNPs were formulated with 6Ac1-C12/DOTAP/DMG-PEG2000 and 6Ac1-C12/DDAB/DMG-PEG2000 at the molar ratio of 60/60/1.2 and 50/40/0.2, respectively. 6Ac1-C12/mRNA weight ratio was set as 20/1. For preparation of 4-Comp Lung LNPs, 6Ac1-C12/DOTAP/cholesterol/DMG-PEG2000 and 6Ac1-C12/DDAB/cholesterol/DMG-PEG2000 at the molar ratio of 60/60/40/1.2 and 50/40/50/0.2 were used. 5-Comp Lung LNPs were prepared as per the SORT strategy[12], supplying 50% (mol) permanently cationic lipids to previous liver-targeted 6Ac1-C12 LNP formulations. In addition, DLin-MC3-DMA, SM-102, and ALC-0315 lipids were utilized to validate the universal applicability of our lung-targeting strategies, and ionizable lipid/DOTAP/DMG-PEG2000 molar ratio of 60/60/1.2 were used here.

### LNP size and stability characterization
For LNP size and stability measurement, nAcx-Cm Liver and Lung LNPs were evaluated. Liver LNPs were formulated with nAcx-Cm/DOPE/cholesterol/DMG-PEG2000 molar ratio of 15/20/25/2. 3-Comp Lung LNPs were formulated with 6Ac1-C12/DOTAP/DMG-PEG2000 molar ratio of 60/60/1.2. 4-Comp Lung LNPs were formulated with 6Ac1-C12/DOTAP/cholesterol/DMG-PEG2000 molar ratio of 60/60/40/1.2. 5-Comp Lung LNPs were prepared as per the SORT strategy[12], supplying 50% (mol) DOTAP to 6Ac1-C12 Liver LNPs. Sizes of these LNPs were measured at different time intervals.

### In vivo accumulation of LNP/Cy5-mRNA formulations
6Ac1-C12 Liver LNPs were formulated at 6Ac1-C12/DOPE/cholesterol/DMG-PEG2000 molar ratio of 15/20/25/2. For 6Ac1-C12 or SM-102 Lung LNPs, 3-, 4-, and 5-Comp Lung LNPs were formulated at ionizable lipid/DOTAP/DMG-PEG2000 molar ratio of 60/60/1.2, ionizable lipid/DOTAP/cholesterol/DMG-PEG2000 molar ratio of 60/60/40/1.2, and ionizable lipid/DOPE/cholesterol/DMG-PEG2000/DOTAP molar ratio of 15/20/25/2/62, respectively. These LNPs encapsulating Cy5-Fluc mRNA (0.5 mg kg⁻¹) were i.v. injected into C57BL/6 mice. 2 h post adminstration, mice were sacrificed and organs were isolated. The fluorescences of organs were imaged using the IVIS Spectrum system.

### In vivo toxicity assay
In vivo toxicity of representative liver- and lung-targeted LNPs were evaluated. 5Ac1-C12 and 6Ac1-C12 Liver LNPs were formulated at the ionizable lipid/DOPE/cholesterol/DMG-PEG2000 molar ratio of 15/20/

25/2. 3-, 4-, and 5-Comp Lung LNPs were formulated at 6Ac1-C12/DOTAP/DMG-PEG2000 molar ratio of 60/60/1.2, 6Ac1-C12/DOTAP/cholesterol/DMG-PEG2000 molar ratio of 60/60/40/1.2, and 6Ac1-C12/DOPE/cholesterol/DMG-PEG2000/DOTAP molar ratio of 15/20/25/2/62, respectively. All LNPs were i.v. injected into C57BL/6 mice at a mRNA dose of higher than that required for high protein expression (0.5 mg kg$^{-1}$). Lipopolysaccharide (LPS, 5 mg kg$^{-1}$, intraperitoneally) and 1× PBS (i.v.) were set as the positive and negative control, respectively. 24 h post administration, the whole blood of the C57BL/6 mice was collected into BD Microtainer tubes and the serum was separated. Liver function including alanine aminotransferase (ALT) and aspartate aminotransferase (AST), and renal function including blood urea nitrogen (BUN) and creatinine (CREA) were evaluated in Hangzhou Liangying Technology Co. Tissue sections including heart, liver, spleen, lung, and kidney, and H&E staining were performed in Wuhan Sevier Biotechnology Co. Slides were scanned using Pannorsmic DESK software (3DHISTECH) and analyzed using CaseViewer software (version 2.4, 3DHISTECH). Hematological analysis was evaluated by Automatic Blood Analyzer (TECOM) in Hangzhou Haoke Biotechnology Co.

### Cell type analysis of mRNA translation with flow cytometry

Flow cytometry was used to identify tdTomato positive cells in different cell types of livers and lungs. 5Ac1-C12 or 6Ac1-C12 Liver LNPs were formulated at ionizable lipid/DOPE/cholesterol/DMG-PEG2000 molar ratio of 15/20/25/2. For 6Ac1-C12 Lung LNPs, 3-, 4-, and 5-Comp Lung LNPs were formulated at ionizable lipid/DOTAP/DMG-PEG2000 molar ratio of 60/60/1.2, ionizable lipid/DOTAP/cholesterol/DMG-PEG2000 molar ratio of 60/60/40/1.2, and ionizable lipid/DOPE/cholesterol/DMG-PEG2000/DOTAP molar ratio of 15/20/25/2/62, respectively. Ai9 mice were i.v. treated with these LNPs loading Cre mRNA (0.25 mg kg$^{-1}$). Cell isolation and staining were conducted 48 h post administration.

To isolate mouse liver cells, separation was conducted by differential centrifugation[57]. Mice were anesthetized with isoflurane and fixed. The mouse liver was perfused, and then cut and digested with collagenase IV (5 mL) at 37 °C for 30 min. After terminating the digestion, the liver cells were passed through a 70 μm cell filter and washed with 1× PBS. Liver parenchyma cells were collected by centrifugation (50 g, 5 min), and the cell precipitate was resuspended in the washing medium and washed with 1× PBS. The supernatant was collected by centrifugation (450 g, 5 min) to give liver non-parenchymal cells. These cells were resuspended in the red blood lysis buffer and incubated for 5 min, then 1× PBS was added to terminate the lysis. The mixture was centrifuged, counted, and resuspended in the cell staining buffer. Following this, staining antibodies were added, and the cells were incubated on ice for 30 min in the dark. The cells were then washed twice with 1× PBS and finally resuspended in 500 μL of 1× PBS. The cell suspension was transferred to the flow tubes and analyzed using a multicolor analytical flow analyzer (LSR Fortessa, BD Biosciences). Antibodies used here included PerCP/Cyanine5.5 anti-mouse CD45 (1/200 dilution, Biolegend, 157208), PE/Cyanine7 anti-mouse CD31 (1/200 dilution, Biolegend, 102524), FITC anti-mouse/human CD11b (1/100 dilution, Biolegend, 101205), and APC anti-mouse F4/80 (1/100 dilution, Biolegend, 123116). Sytox™ Blue Dead Cell Stain (1/2000 dilution, Thermo Fisher, S34857) was used to differentiate the live cells.

To isolate and stain lung cells, the lung was minced and added to a 15 mL tube containing Collagenase I and DNase I. The mixture was incubated at 37 °C for 1 h. Following termination of digestion, the mixture was filtered through a 70 μm cell filter and washed with 1× PBS. The remaining steps were consistent with the protocol of liver cell collection. Antibodies used here included APC anti-mouse CD45 (1/200 dilution, Biolegend, 103112), PE/Cyanine7 anti-mouse CD31 (1/200 dilution, Biolegend, 102524), and FITC anti-mouse CD326 (Ep-CAM) (1/100 dilution, Biolegend, 118207). Sytox™ Blue Dead Cell Stain (1/2000 dilution,

Thermo Fisher, S34857) was used to differentiate the live cells. Ultimately, lung cells were analyzed using the LSRForessa machine. Data were analyzed using FLOWJO software version 10.8 (FLOWJO).

### Statistical analyses

Statistical analyses were conducted with GraphPad Prism 9. Data were reported as mean±s.d. The statistical analysis methods were described in the figure legends. A two-tailed un-paired Student's t-test and a one-way analysis of variance (ANOVA) was performed when comparing two groups and multiple groups, respectively. *P-values < 0.05, **P < 0.01, ***P < 0.001, and ****P < 0.0001 were considered statistically significant.

### Reporting summary

Further information on research design is available in the Nature Portfolio Reporting Summary linked to this article.

## Data availability

All data supporting the findings of this study are presented in the Article, Supplementary Information, and Source Data file. The data that support the findings of this study are available from the corresponding author upon reasonable request. Source data are provided with this paper. Source data is available for Figs. 2–5 and Supplementary Figs. 3–7, 9, 17, 19, 22, 25, 26, 31, 32 and 34 in the associated source data file. Source data are provided with this paper.

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

## Acknowledgements

This work was supported by the National Natural Science Foundation of China (Grant No. 22205201 to S.L.), the Zhejiang Provincial Natural Science Foundation of China (Grant No. LZ23H180003 to S.L.), and the Startup Package of Zhejiang University (to S.L.). We thank Jianyang Pan (Research and Service Center, College of Pharmaceutical Sciences, Zhejiang University) for performing NMR spectrometry for structure elucidation. We thank Lingyun Wu and Tingyu Liu in the Center of Cryo-Electron Microscopy (CCEM), Zhejiang University for their assistance on Cryo-Transmission Electron Microscopy. We thank Bin Xu and Shichun Shao for the technical support by the Core Facilities, Liangzhu Laboratory.

## Author contributions

K.S. and S.L. designed the experiments and wrote the manuscript. S.L. supervised the project. K.S., L.S., T.S., X.Y., L.L, C.M., S.W., Y.C., Y.Z., C.W., Z.W., J.Q., J.Z. and T.X. performed the experiments and analyzed the data. Y.P., Z.G. and S.L. conceived the idea. All authors discussed the results and commented on the manuscript.

## Competing interests

K.S. and S.L. have filed the patent application on this research. Y.P. is a scientific co-founder of Ruidax. Z.G. is the co-founder of Zenomics Inc., Zcapsule Inc. and μZen Inc. The other authors declare no competing interests.
