## [Peer Review File · Nature Communications]

Reformulating lipid nanoparticles for organ-targeted mRNA accumulation and translationREVIEWER COMMENTS

Reviewer #1 (Remarks to the Author):

The authors developed a combinatorial library of cationic lipids with degradable ester cores to identify novel lipids for mRNA delivery via LNPs. They show mRNA delivery and expression in an ovarian cancer cell line and in mice using LNPs that contain their modified lipids. They also show mRNA delivery and translation using LNPs without cholesterol and phospholipids. This latter finding is interesting as standard LNP formulations are known to be comprised of 4 essential components: ionizable lipids, cholesterol, phospholipids, PEG-lipid. It is unclear if cholesterol and phospholipids become unnecessary because of the new modified lipids developed by the authors. Overall, there are some potential interesting findings. However, the text and figures lack clarity and additional controls are needed to support the claims (as specified below).

1. For the Rational design section starting at line 88, the description is insufficient to explain why the authors think that "altering the intrinsic formula and introducing degradable cores into cationic lipids might provide a way forward for...next-generation lipid vectors with high efficacy and safety." Why do the authors think that a degradable core will improve mRNA delivery and expression? Please walk us through your logic. It is difficult to follow the way it is currently written.

2. The authors should explain the nomenclature for nAcx-Cm in the body of the paper. The explanation is buried in the Fig. 1 caption.

3. Legend needed for Fig. 1e left image to convey composition of standard LNPs.

4. Where are images from e and f from? Clarification is needed in the caption.

5. It's unclear why the authors are using the term "veritable targeting" so frequently through the paper. Please clarify.

6. Clarification for Fig. 2a caption is needed. What is "effect trend"?

7. Labels are needed for organs in Fig. 2c and 4a. Unclear which organs are lighting up (expressing the mRNA).

8. Why are LNPs optimized for ovarian cancer cell line transfection if the goal is lung/liver mRNA delivery? (line 116)

9. Why is SM-102 and ALC-0315 LNPs only used in the comparison of intramuscular injection, while DLin-MD3-DMA LNPs are used in comparison for intravenous injection? Please clarify.

10. Kupffer cells in the liver are known to take up nanoparticles non-specifically. How can the

authors claim this is targeted mRNA delivery? (lines 312-322)

11. The authors claim that the absence of cholesterol allows mRNA delivery to liver endothelial cells rather than hepatocytes. However, a head-to-head comparison of their modified LNPs with standard cholesterol-containing LNPs in Fig. 5c is missing to show the altered mRNA distribution amongst the Kupffer, endothelial, and hepatocyte cells.

Reviewer #2 (Remarks to the Author):

The work is focusing on introduction of 3 components approach where ionizable lipids replaces cholesterol and phospholipids. The targeting more precisely of specific organ in this case is mainly either liver or lung is mainly modulated by removal of cholesterol and in most cases usage of permanently positively charged lipid. This work is inspired by Daniel Sigwart papers pioneers in the field of LNPs targeting.

Authors introduced biodegradable ionizable lipids as a base of the formulation and structure activity relationship was studied extensively. The conclusions are mainly supported by data. However, there are some points are missing, and it is clarified in the attached file.

Since organs targeted in this paper are mainly liver and lung, the title "Reformulating lipid nanoparticles for simultaneous organ-targeted mRNA accumulation and translation" is too ambitious. It should be more focus on precision of targeting of liver and lung rather than organs in general (even though in one instance there was accumulation in spleen too). Additional detail that should be addressed is the repeated usage of the expression "simultaneous accumulation and translation" is confusing since translation can happen only after accumulation in some location. Methodology, details provided are acceptable and are in line with the standard with similar works in the field.

The paper may be accepted after addressing comments.

The paper presented an impressive amount of work overall. The manuscript focuses on using biodegradable ionizable lipids in LNP formulations targeting specific organs by utilizing a 3 three-component approach to eliminating phospholipid and cholesterol. The article extensively studies the structure and activity relationship of branched biodegradable ionizable lipids. The targeting shown in the manuscript focused mainly on the liver and lungs.

I have the following questions:

1. Line 200: "Additionally, nAcxCm LNPs exhibited pKa values around 6.0 (Fig. 3j). These characteristics suggest that nAcx2 Cm LNPs possess excellent stability before cellular entry and mRNA can be rapidly escaped from endosomes and released into the cytoplasm for subsequent translation."

Why does PKa around 6 suggest that LNPs were stable before entry? Please clarify.

2. Figure 2b: Please provide in the table n/p ratios.
3. Did you test your ionizable lipids as a two-component formulation, just ionizable lipid, and DMG-PEG? If positive, did you see transfection in vitro and in vivo, and which organs were transfected?
4. Figure 3 a: please provide encapsulation efficiency for the formulations.
5. Line 269: “Notably, the simplified 3-Comp Lung LNPs facilitated higher efficacy compared to their 4-Comp or 5-Comp counterparts, highlighting the significance of LNP engineering innovation for superior efficacy and targeting (Supplementary Fig. 18)”.

Did you compare your 3, 4, and 5-component formulations in vitro? If yes, please provide the data. If not, please demonstrate transfection in vitro. It will be interesting to see the in-vitro in-vivo correlation for the innovative 3-component approach.

6. Line 351: “Moreover, these Liver—or Lung-targeted nAcx-Cm LNPs were well tolerated in vivo.” The authors demonstrated some data for liver enzymes, kidney BUN, and creatinine, as well as histology for the liver and lungs. A total blood count should be performed for the three-component formulation to confirm this conclusion.

Reviewer #3 (Remarks to the Author):

In this study, for organ-specific targeting of mRNA-encapsulated lipid nanoparticles, the authors focused on lipid composition and found a new 3-Comp LNP strategy with cholesterol removed to eliminate accumulation in the liver, which I believe is novel enough. The mRNA delivery to organs other than the liver is an important issue, and we believe that we have found important findings for the future development of mRNA drugs. The experimental methods and results for drawing conclusions are also reasonable. However, there is insufficient citation of previous studies on cholesterol content in the lipid composition of mRNA-encapsulated lipid nanoparticles. Therefore, prior studies on the effect of lipid composition of mRNA-encapsulated lipid nanoparticles should be cited and well discussed in order to be accepted.

Response to reviewers:

Reviewer #1 (Remarks to the Author):

Comments:

The authors developed a combinatorial library of cationic lipids with degradable ester cores to identify novel lipids for mRNA delivery via LNPs. They show mRNA delivery and expression in an ovarian cancer cell line and in mice using LNPs that contain their modified lipids. They also show mRNA delivery and translation using LNPs without cholesterol and phospholipids. This latter finding is interesting as standard LNP formulations are known to be comprised of 4 essential components: ionizable lipids, cholesterol, phospholipids, PEG-lipid. It is unclear if cholesterol and phospholipids become unnecessary because of the new modified lipids developed by the authors. Overall, there are some potential interesting findings. However, the text and figures lack clarity and additional controls are needed to support the claims (as specified below).

Response: We sincerely thank the reviewer for thoroughly reviewing our manuscript. Typically, standard LNPs are comprised of ionizable lipids, cholesterol, phospholipids, and PEG-lipid. In this manuscript, cholesterol and phospholipid were still required for formulating Liver-targeted LNPs. Encouragingly, our ionizable lipid/permanently cationic lipid/PEG-lipid 3-component (3-Comp) LNPs, which exclude cholesterol and phospholipids, facilitated high mRNA accumulation and translation in the lung (**Fig. 4f-h** and **5i**). Simultaneously, these 3-Comp LNPs exhibited excellent stability after long-term storage (**Fig. 5n**). These results demonstrate that cholesterol and phospholipids become unnecessary in our 3-Comp Lung LNPs.

1. For the Rational design section starting at line 88, the description is insufficient to explain why the authors think that "altering the intrinsic formula and introducing degradable cores into cationic lipids might provide a way forward for...next-generation lipid vectors with high efficacy and safety." Why do the authors think that a degradable core will improve mRNA delivery and expression? Please walk us through your logic. It is difficult to follow the way it is currently written.

Discussion: We sincerely thank the reviewer for this comment. Previous cationic lipid architectures typically originate from amines, featuring an amine-linker-tail formula. In some lipid cases, the linkers contain degradable groups. However, even if the linkers may cleave, only hydrophobic tails are detached from the ionizable lipids, leaving behind amine and residues in the resulting relatively large molecules. Conversely, if multiple degradable sites exist in the lipid core, comprehensive core degradation will lead to smaller moieties. Also, potential toxic amines will be cleaved to several small residues. This thorough degradation will efficiently facilitate the LNP dissociation, mRNA release, and improve safety profiles. Therefore, we believe that the degradable core will enhance the mRNA delivery and expression. As expected, our core-degradable ionizable lipid based LNPs achieved rapid nanoparticle dissociation, efficient mRNA release, good safety profiles, and highly efficacious mRNA delivery (**Fig. 3h,i, 4a,b,f-h, 5i,o** and **Supplementary Figs. 32-34**).

Action: We have further added the explanation "However, even if the linkers may cleave, only hydrophobic tails are detached from the ionizable lipids, leaving behind amine and residues in the resulting molecules. Conversely, comprehensive core degradation of lipids will lead to smaller

moieties, thereby facilitating LNP dissociation, mRNA release, and safety profiles.” in the main text (Page 3). We have labeled the added sentences with yellow highlights.

2. The authors should explain the nomenclature for nAcx-Cm in the body of the paper. The explanation is buried in the Fig. 1 caption.

Discussion: We thank the reviewer for this suggestion and we agree. In the nomenclature ‘nAcx-Cm’, ‘Ac’ indicated acrylic esters, ‘n’ denoted the number of ester bonds in one molecule, and ‘m’ referred to different alkyl chains in the lipids.

Action: We have added a more detailed explanation of the nomenclature for nAcx-Cm in the main text (Page 3). The added sentences were labeled with yellow highlights.

3. Legend needed for Fig. 1e left image to convey composition of standard LNPs.

Discussion: We thank the reviewer for carefully checking our manuscript. The reason we did not label the LNP composition in the original Fig. 1e is that there are multiple strategies for achieving lung-targeted delivery, and the composition is not consistent across these methods. However, we agree with the suggestion to label the commonly used compositions in the left image of Fig. 1e. This addition will make our schematic clearer. Thank you.

Action: We have supplied the new Fig. 1e.

4. Where are images from e and f from? Clarification is needed in the caption.

Discussion: We thank the reviewer for this helpful comment.

Action: We have added “Representative mRNA accumulation and translation images from 6Ac1-C12 4-Comp Lung LNPs (imperfect lung targeting), 6Ac1-C12 3-Comp Lung LNPs (veritable lung

targeting), and 6Ac1-C12 Liver LNPs (conventional compositions, veritable liver targeting) were presented, respectively” in the caption of **Fig. 1e,f** (Page 4). We have labeled the added sentences with yellow highlights.

5. It's unclear why the authors are using the term "veritable targeting" so frequently through the paper. Please clarify.

Response: We thank the reviewer for this comment. The current mRNA targeted delivery remains incomplete and non-veritable. Existing targeting systems exclusively enable organ-selective mRNA translation, while do not achieve organ-targeted accumulation following systemic administration. Despite some progress in extrahepatic mRNA translation, a substantial proportion of nanoparticles still distribute in the liver (e.g. our *Nat. Mater.* **20**, 701-710 (2021); *Proc. Natl Acad. Sci. USA* **118**, e2109256118 (2021); and *Proc. Natl Acad. Sci. USA* **119**, e2116271119 (2022)). Veritable targeting should achieve simultaneous organ-targeted mRNA accumulation and translation, which remains a formidable challenge. Encouragingly, our 3-Comp Lung LNPs achieved lung-targeted mRNA accumulation and translation. This represents a significant innovation in the mRNA delivery field, so we frequently used the term "veritable targeting" in this manuscript.

6. Clarification for Fig. 2a caption is needed. What is "effect trend"?

Discussion: We thank the reviewer for this helpful comment. The “effect trend” is based on the orthogonal screening method, demonstrating variations in mRNA expression when the molar ratios of different LNP components are changed. The effect trend enables further optimization for the best formulation.

Action: Since **Fig. 2a** did not show "effect trend" result, we removed “and effect trend of each component” from the caption. Instead, we supplied more "effect trend" data and clarification in the **new Supplementary Fig. 3**. As shown below, we can observe the trend between mRNA delivery efficacy and the content of each lipid.

Supplementary Fig. 3 | The first-round orthogonal screening of the four components' molar

ratios. The optimization of the formulation was based on four levels and the effect trend of each LNP component was shown.

7. Labels are needed for organs in Fig. 2c and 4a. Unclear which organs are lighting up (expressing the mRNA).

Discussion: We thank the reviewer for this comment.

Action: We have revised **Fig. 2c** and **4a**, and labeled the organs in the images. In addition, we defined the labels in the caption. The added sentences were labeled with yellow highlights (Page 5 and 8).

8. Why are LNPs optimized for ovarian cancer cell line transfection if the goal is lung/liver mRNA delivery? (line 116)

Response: We thank the reviewer for carefully checking our manuscript. Due to additional barriers for in vivo delivery, numerous vectors with in vitro activity fail to translate to animal models (our *Nat. Mater.* **20**, 701–710 (2021); *Angew. Chem. Int. Ed.* **55**, 12013–12017 (2016); and others). There is often inconsistency between in vitro and in vivo results, making it challenging to develop organ-targeted mRNA delivery systems based solely on in vitro evaluation. Therefore, we used the ovarian cancer cell line for initial in vitro screening to pursue potentially effective vectors, but focused more on in vivo delivery to achieve organ-targeting. The ovarian cancer cell line was widely utilized in our previous work (e.g. our *Nat. Mater.* **20**, 701–710 (2021); our *J. Am. Chem. Soc.* **143**, 21321–21330 (2021); our *Angew. Chem. Int. Ed.* e202405444 (2024) doi:10.1002/anie.202405444; and others).

9. Why is SM-102 and ALC-0315 LNPs only used in the comparison of intramuscular injection, while DLin-MC3-DMA LNPs are used in comparison for intravenous injection? Please clarify.

Discussion: We thank the reviewer for this comment. SM-102 and ALC-0315 LNPs are used in two FDA-approved mRNA vaccines (mRNA-1273 and BNT162b2) and are intramuscularly injected. DLin-MC3-DMA LNPs are intravenously administered in the FDA-approved siRNA drug (Onpattro). Therefore, our LNPs were compared to SM-102 and ALC-0315 LNPs by intramuscular injection, and DLin-MC3-DMA LNPs were used in comparison for intravenous injection. In the revised version, to better exhibit the high performance of 6Ac1-C12 LNPs, we complemented all these positive controls with both intramuscular and intravenous administration.

Action: In the previous version, 6Ac1-C12 LNPs outperformed DLin-MC3-DMA LNPs post intravenous injection, and showed higher efficacy compared to SM-102 and ALC-0315 LNPs after intramuscular administration (**Fig. 4a,b**). We complementally further compared the mRNA delivery efficiency of 6Ac1-C12, SM-102, and ALC-0315 LNPs by intravenous injection. 6Ac1-C12 LNPs exhibited higher or comparable mRNA delivery efficacy than SM-102 and ALC-0315 LNPs. In addition, with intramuscular injection, 6Ac1-C12 LNPs showcased superior delivery capacity compared to DLin-MC3-DMA LNPs (**new Supplementary Fig. 11**). We have revised the main text and labeled added and/or revised sentences with yellow highlights (Page 7).

Supplementary Fig. 11 | The comparison of mRNA delivery efficiency of the optimal lipid 6Ac1-C12 with commercialized lipids by different administration routes. a, 6Ac1-C12 LNPs exhibited higher or comparable Fluc mRNA expression than SM-102 and ALC-0315 LNPs via i.v. injection. **b**, C57BL/6 mice were treated with LNPs containing Fluc mRNA and bioluminescence was quantified 6 h post intramuscular injection. 6Ac1-C12 LNPs outperformed SM-102, ALC-0315, and DLin-MC3-DMA LNPs by intramuscularly delivering mRNA.

10. Kupffer cells in the liver are known to take up nanoparticles non-specifically. How can the authors claim this is targeted mRNA delivery? (lines 312-322)

Response: We thank the reviewer for carefully checking our manuscript. We agree that the uptake of nanoparticles by Kupffer cells is non-specific. Our proposed strategy aims to achieve targeted mRNA delivery at the organ (liver or lung) level, rather than at the specific cell type level. Achieving targeted mRNA delivery to specific cell types remains a global challenge, and our team will continue to work towards developing cell type targeted delivery systems.

11. The authors claim that the absence of cholesterol allows mRNA delivery to liver endothelial cells rather than hepatocytes. However, a head-to-head comparison of their modified LNPs with standard cholesterol-containing LNPs in Fig. 5c is missing to show the altered mRNA distribution amongst the Kupffer, endothelial, and hepatocyte cells.

Response: We thank the reviewer for this comment. There may have been some misunderstandings. We did not claim that the delivery of mRNA to liver endothelial cells rather than hepatocytes was due to the absence of cholesterol. In **Fig. 5a-e**, 6Ac1-C12 Liver LNPs were formulated with 6Ac1-C12/DOPE/Cholesterol/DMG-PEG2000, inducing mRNA delivery to liver endothelial cells. The endothelial cell tropism might be attributed to new ionizable lipid structures. We previously included “Starting from the typical helper lipids comprising DOPE, cholesterol, and PEG-lipid, 6Ac1-C12 LNPs enabled liver-targeted accumulation and protein expression (Figs. 4c and 5a,b)” in the main text (Page 10) to avoid misunderstanding. The absence of cholesterol applied to our 3-Comp Lung LNPs, facilitating simultaneous lung-targeted mRNA accumulation and translation.

Again, we thank reviewer 1 for carefully checking our manuscript and providing helpful comments to improve our manuscript.

Reviewer #2 (Remarks to the Author):

Comments:

The work is focusing on introduction of 3 components approach where ionizable lipids replaces cholesterol and phospholipids. The targeting more precisely of specific organ in this case is mainly either liver or lung is mainly modulated by removal of cholesterol and in most cases usage of permanently positively charged lipid. This work is inspired by Daniel Sigwart papers pioneers in the field of LNPs targeting.

Authors introduced biodegradable ionizable lipids as a base of the formulation and stricture activity relationship was studied extensively. The conclusions are mainly supported by data. However, there are some points are missing, and it is clarified in the attached file.

Since organs targeted in this paper are mainly liver and lung, the title "Reformulating lipid nanoparticles for simultaneous organ-targeted mRNA accumulation and translation" is too ambitious. It should be more focus on precision of targeting of liver and lung rather than organs in general (even though in one instance there was accumulation in spleen too). Additional detail that should be addressed is the repeated usage of the expression "simultaneous accumulation and translation" is confusing since translation can happen only after accumulation in some location.

Methodology, details provided are acceptable and are in line with the standard with similar works in the field.

The paper may be accepted after addressing comments.

(Please see attached)

Discussion: We thank the reviewer for carefully checking our manuscript and providing insightful comments. We agree that the title is ambitious. We have revised the title to “Reformulating lipid nanoparticles for liver- and lung-targeted mRNA accumulation and translation”.

mRNA translation can happen only after accumulation in some location. However, the occurrence of lung-targeted mRNA translation does not necessarily imply exclusive accumulation in the lung. Numerous existing targeting systems exclusively enable organ-selective mRNA translation, while do not achieve organ-targeted accumulation following systemic administration. Despite the progress in extrahepatic mRNA translation, a substantial proportion of nanoparticles still distribute in the liver (e.g. our Nat. Mater. **20**, 701-710 (2021); Proc. Natl Acad. Sci. USA **118**, e2109256118 (2021); and Proc. Natl Acad. Sci. USA **119**, e2116271119 (2022)). Encouragingly, our conventional four-component LNPs mediated liver-targeted mRNA accumulation and translation, and our 3-Comp Lung LNPs resulted in targeted mRNA accumulation and translation in the lung. Because of this innovation, we repeated used the "simultaneous accumulation and translation" in our manuscript.

Action: We have revised the title to “Reformulating lipid nanoparticles for liver- and lung-targeted mRNA accumulation and translation”. We have revised the main text and labeled added and/or revised sentences with yellow highlights (Page 1).

The paper presented an impressive amount of work overall. The manuscript focuses on using biodegradable ionizable lipids in LNP formulations targeting specific organs by utilizing a 3 three-component approach to eliminating phospholipid and cholesterol. The article extensively studies the structure and activity relationship of branched biodegradable ionizable lipids. The targeting shown in the manuscript focused mainly on the liver and lungs.

I have the following questions:

1. Line 200: “Additionally, nAcx-Cm LNPs exhibited pKa values around 6.0 (Fig. 3j). These characteristics suggest that nAcx-Cm LNPs possess excellent stability before cellular entry and mRNA can be rapidly escaped from endosomes and released into the cytoplasm for subsequent translation.” Why does pKa around 6 suggest that LNPs were stable before entry? Please clarify.

Discussion: We thank the reviewer for this comment. It seems there may have been a misunderstanding due to the lack of clarity in our statement. We would like to clarify that in this context, these multiple physicochemical properties (e.g. sizes, endosomal escape of nanoparticles, and mRNA release capacity) presented in **Fig. 3** collectively contribute to the stability, endosomal escape, and mRNA release of nAcx-Cm LNPs, rather than pKa.

Action: To avoid misunderstanding, we have added the figure ranges (Fig. 3b-i) after this sentence and highlighted it in yellow in the main text (Page 7). Thank you for pointing out this.

2. Figure 2b: Please provide in the table n/p ratios.

Discussion: We thank the reviewer for this helpful comment and we agree.

Action: We have supplemented the N/P ratios in the table (**revised Fig. 2b**).

b

	Factors (molar ratio)				6Ac1-C12/mRNA (w/wt)	N/P
	6Ac1-C12	DOPE	Cholesterol	DMG-PEG2000		
A-2	10	15	20	2	10	11
A-3	10	20	25	3	10	11
A-4	10	25	30	4	10	11
A-5	15	10	20	3	10	11
A-8	15	25	25	2	10	11
A-12	20	25	20	1	10	11
B-1	15	30	25	2	10	11
B-2	15	20	25	2	10	11
B-3	15	15	25	2	10	11
B-4	15	20	25	2	20	22

Fig. 2b | LNP formulation table for mRNA delivery validation in vivo.

3. Did you test your ionizable lipids as a two-component formulation, just ionizable lipid, and DMG-PEG2000? If positive, did you see transfection in vitro and in vivo, and which organs were transfected?

Discussion: We thank the reviewer for this comment.

Action: We proceeded to test the two-component formulation (containing ionizable lipids and DMG-PEG2000) as suggested. Unfortunately, the two-component LNPs did not achieve efficient mRNA delivery. The data were shown in **new Supplementary Fig. 16**. We have revised the main text and labeled added and/or revised sentences with yellow highlights (Page 9).

Supplementary Fig. 16 | In vivo quantification of Fluc mRNA delivery via 6Ac1-C12/DMG-PEG2000 two component LNPs. 2.5% or 5% (wt) DMG-PEG2000 was used here. The two component LNPs could not enable efficacious mRNA delivery in vivo.

4. Figure 3a: please provide encapsulation efficiency for the formulations.

Discussion: We thank the reviewer for this comment. Previously, we provided the mRNA encapsulation efficiency data for 5Ac1-C12, 6Ac1-C12, and 6Ac1-C212 LNPs. We agree that additional data on the encapsulation efficiency of representative candidates are necessary.

Action: To address this concern, we further measured the encapsulation efficiency of more representative nAcx-Cm LNPs, and the data were presented in **new Supplementary Fig. 9c**. In addition, we tested the mRNA binding (encapsulation) efficacy of 6Ac1-C12 Lung LNPs with different components and the data were showed in **new Supplementary Fig. 17c**.

Supplementary Fig. 9c | mRNA binding efficacy (c) of representative candidates were evaluated. Data are presented as mean ± s.d. (n = 3 biologically independent samples).

Supplementary Fig. 17c |mRNA binding (c)..... were evaluated. Data are presented as mean \pm s.d. (n = 3 biologically independent samples).

5. Line 269: “Notably, the simplified 3-Comp Lung LNPs facilitated higher efficacy compared to their 4-Comp or 5-Comp counterparts, highlighting the significance of LNP engineering innovation for superior efficacy and targeting (Supplementary Fig. 18)”.

Did you compare your 3, 4, and 5-component formulations in vitro? If yes, please provide the data. If not, please demonstrate transfection in vitro. It will be interesting to see the in-vitro in-vivo correlation for the innovative 3-component approach.

Discussion: We thank the reviewer for this helpful comment. Exploring the correlation between in vitro and in vivo mRNA delivery efficacy is indeed valuable. While we evaluated both in vitro and in vivo delivery capabilities, we did not observe a clear correlation between them.

Action: To assess the correlation between in vitro and in vivo mRNA delivery efficacy, we conducted the in vitro transfection for the 3-, 4-, and 5-Comp LNPs, and the data were presented in the Supplementary information (**new Supplementary Fig. 17b**). Interestingly, although 3-Comp Lung LNPs outperformed 4- and 5-Comp LNPs for mRNA delivery in vivo, they mediated lower mRNA expression in vitro. This is a common phenomenon, as inconsistency often exists between in vitro and in vivo results (e.g. our *Nat. Mater.* **20**, 701–710 (2021); *Angew. Chem. Int. Ed.* **55**, 12013–12017 (2016); and others).

Supplementary Fig. 17b |in vitro mRNA delivery efficacy (b).....were evaluated. Data are presented as mean \pm s.d. (n = 3 biologically independent samples).

6. Line 351: “Moreover, these Liver—or Lung-targeted nAcx-Cm LNPs were well tolerated in vivo.” The authors demonstrated some data for liver enzymes, kidney BUN, and creatinine, as well as histology for the liver and lungs. A total blood count should be performed for the three-component formulation to confirm this conclusion.

Discussion: We thank the reviewer for this helpful comment. We agree that evaluating a total blood count would further demonstrate the tolerance of nAcx-Cm LNPs in vivo.

Action: To further verify the low toxicity of nAcx-Cm LNPs in vivo, we conducted the hematological analysis. The data of major hematological analysis were added in the Supplementary Information (**new Supplementary Fig. 34**). The hematological analysis of the mice administrated with nAcx-Cm LNPs did not show significant differences compared to the PBS negative control group. The results demonstrate that nAcx-Cm LNPs with different components are well tolerated in vivo.

Supplementary Fig. 34 | nAcx-Cm Liver and Lung LNPs were well tolerated in vivo. nAcx-Cm Liver and Lung LNPs were i.v. administrated to C57BL/6 mice. PBS (i.v.) was set as the negative control. **a**, Major hematological analysis was performed 24 h post administration. The main hematological analysis included assessments for white blood cell (WBC), neutrophil (NEU), lymphocyte (LYM), monocyte (MON), red blood cell (RBC), platelet (PLT), mean corpuscular volume (MCV), hemoglobin (HGB), and hematocrit (HCT). There was no significant difference between the nAcx-Cm LNPs and PBS groups. **b**, The major hematological analysis of 3-Comp Lung LNPs showed no significant differences compared to the negative control. All data are presented as mean \pm s.d. (n=3 biologically independent animals). Statistical significances in **a** were calculated using one-way ANOVA with Dunnett’s multiple comparisons test, and those in **b** were calculated with a two-tailed unpaired t-test: **** $P < 0.0001$; *** $P < 0.001$; ** $P < 0.01$; * $P < 0.05$; ns, no significant difference.

Again, we thank reviewer 2 for carefully checking throughout our manuscript and providing insightful comments for revisions.

Reviewer #3 (Remarks to the Author):

Comments:

In this study, for organ-specific targeting of mRNA-encapsulated lipid nanoparticles, the authors focused on lipid composition and found a new 3-Comp LNP strategy with cholesterol removed to eliminate accumulation in the liver, which I believe is novel enough. The mRNA delivery to organs other than the liver is an important issue, and we believe that we have found important findings for the future development of mRNA drugs. The experimental methods and results for drawing conclusions are also reasonable. However, there is insufficient citation of previous studies on cholesterol content in the lipid composition of mRNA-encapsulated lipid nanoparticles. Therefore, prior studies on the effect of lipid composition of mRNA-encapsulated lipid nanoparticles should be cited and well discussed in order to be accepted.

Discussion: We thank the reviewer for pointing out this important issue. We agree that cholesterol-removal represents a critical aspect in our mRNA delivery systems, and more cholesterol-related LNP research should be cited and discussed. Although cholesterol is used in FDA-approved LNPs (Onpattro, mRNA-1273, and BNT162b2), its structural modification is still necessary for higher delivery efficacy and organ-targeting achievement. For this purpose, numerous efforts have been made to functionalize cholesterol with chemical or charge variations.

Action: We further cited more references (e.g., *Adv. Funct. Mater.* **33**, 2303795 (2023); *Proc. Natl Acad. Sci. USA* **121**, e2307801120 (2024); *Adv. Sci.* **10**, 2300188 (2023); *Nat. Commun.* **11**, 983 (2020); and *Adv. Mater.* **31**, 1807748 (2019)) and discussed the progress of cholesterol content in LNPs. We have revised the main text and labeled added and/or revised sentences with yellow highlights (Page 10).

We would like to thank reviewer 3 again for helping to improve this manuscript.

REVIEWERS' COMMENTS

Reviewer #2 (Remarks to the Author):

The authors thoroughly addressed all comments and provided all crucial theoretical and experimental data. I recommend accepting the manuscript.

Response to reviewers:

Reviewer #2

The authors thoroughly addressed all comments and provided all crucial theoretical and experimental data. I recommend accepting the manuscript.

Response: We thank the reviewer for checking our manuscript. Thank you for the positive feedback and support of our work.